# Chemical Composition and Antimicrobial and Antioxidant Activities of Essential Oil of Sunflower (*Helianthus annuus* L.) Receptacle

**DOI:** 10.3390/molecules25225244

**Published:** 2020-11-11

**Authors:** Xin-Sheng Liu, Bo Gao, Xin-Lu Li, Wan-Nan Li, Zi-An Qiao, Lu Han

**Affiliations:** 1School of Life Science, Jilin University, Changchun 130012, China; liuxs18@mails.jlu.edu.cn (X.-S.L.); gaobo@jlu.edu.cn (B.G.); xinlu19@mails.jlu.edu.cn (X.-L.L.); liwannan@jlu.edu.cn (W.-N.L.); qiaoza18@mails.jlu.edu.cn (Z.-A.Q.); 2Key Laboratory for Molecular Enzymology and Engineering, Jilin University, Ministry of Education, Changchun 130012, China; 3Key Laboratory for Evolution of Past Life and Environment in Northeast Asia, Jilin University, Ministry of Education, Changchun 130012, China

**Keywords:** sunflower (*Helianthus annuus* L.), essential oil, chemical composition, antimicrobial activity, antioxidant activity

## Abstract

Sunflower (*Helianthus annuus* L.) contains active ingredients, such as flavonoids, alkaloids and tannins. Nevertheless, few studies have focused on essential oil from the receptacle of sunflower (SEO). In this work, we investigated the chemical composition and antimicrobial and antioxidant activities of SEO. The yield of SEO was about 0.42% (*v*/*w*) by hydrodistillation. A total of 68 volatile components of SEO were putatively identified by gas chromatography–mass spectrometry (GC-MS). The main constituents of SEO were α-pinene (26.00%), verbenone (7.40%), terpinolene (1.69%) and α-terpineol (1.27%). The minimum inhibitory concentration (MIC) of SEO against *P. aeruginosa* and *S. aureus* was 0.2 mg/mL. The MIC of SEO against *S. cerevisiae* was 3.2 mg/mL. The MIC of SEO against *E. coli* and *Candida albicans* was 6.4 mg/mL. The results showed that SEO had high antibacterial and antifungal activities. Three different analytical assays (DPPH, ABTS and iron ion reducing ability) were used to determine the antioxidant activities. The results showed that SEO had antioxidant activities. To summarize, the results in this study demonstrate the possibility for the development and application of SEO in potential natural preservatives and medicines due to its excellent antimicrobial and antioxidant activities.

## 1. Introduction

Sunflower (*Helianthus annuus* L.) is an annual dicotyledonous plant that belongs to the family Asteraceae and is widely distributed in North America, Eastern Europe and Northern China [1]. Sunflower roots, stems, leaves and seeds contained phenols, flavonoids and alkaloids [2]. In previous studies, sunflower florets were found to contain dietary fiber and phenolic acid [3]. Sunflower petals were found to have triterpene glycosides, which had anti-inflammatory activity [4]. The ethanolic extract of sunflower seeds has a potential antidiabetic property in type 2 diabetes mellitus [5]. The aqueous extract of sunflower seeds was found to have considerable potential in reducing asthma symptoms [6] and high antioxidant activity [7]. The sunflower receptacle has always been discarded because of a lack of studies focusing on its commercial application.

In previous studies, essential oil was obtained from roots, stems and leaves of plants by hydrodistillation [8]. Significant differences have been found in the essential oil contents of different sunflower parts. Previous research showed that α-pinene was higher in flowers (72.6%) than in leaves (28.6%) [9]. In capitula and pollen, α-pinene was found at 20.0%, in addition to monoterpene and sesquiterpene hydrocarbons [10]. T previous study proved that sunflower head reduced the level of uric acid and thus may be used as a Chinese traditional drug for treating gout [11]. However, the chemical composition and biological activities of sunflower receptacle essential oil (SEO) have not been studied before.

To our knowledge, there are no reports on the antimicrobial and antioxidant activities of SEO. This study aimed to evaluate the chemical composition and antimicrobial and antioxidant activities of SEO to determine the potential application of the sunflower receptacle in food and medical fields. The development and utilization of the sunflower receptacle can benefit from scientific guidance provided through the study of the chemical composition and biological activity of SEO.

## 2. Results and Discussion

### 2.1. Yield and Chemical Composition of SEO

SEO was extracted by hydrodistillation for 8 h. The average yield of SEO was about 0.42% (*v*/*w*) in this study. In the previous study, the yield of essential oil from dried sunflower head was 0.20% (*w/w*) by steam distillation at 100 °C for 6 h [12]. The essential oil yields from the Carlos and Florom 350 hybrid sunflower heads were found to be 0.12% and 0.13% by hydrodistillation for 2 h, respectively [9].

Pentadecane was used as an internal standard, and α-pinene, verbenone, terpinolene and α-terpineol were used as external standards. Sixty eight chemical components of SEO were putatively identified by GC-MS analysis, accounting for 92.07% of the total content of essential oil, as shown in Table 1, Figure A1 and Table A1. The major chemical component was α-pinene (26.00%). The other main components were verbenone (7.40%), calarene (5.27%), kaur-16-ene (3.51%) and terpinolene (1.69%). Among the 68 chemical compounds, most of them were monoterpenoids (54.65%), mainly including α-pinene, α-terpineol, verbenone and terpinolene. Sesquiterpenoids content was 22.73%; content of others was 14.69%.

The main chemical components were α-pinene and verbenone, which were all found in sunflower receptacle, capitula and pollen, as shown in Table 2. Among the main chemical components of essential oil, α-pinene was present in the highest amount. Some chemical components of essential oil, such as terpinolene, α-terpineol, pinocarvone and *cis*-verbenol, were found in both sunflower receptacle and capitula. However, the most of chemical components of essential oil from different sunflower parts (receptacle, capitula and pollen) were different. For example, kaur-16-ene, 16-kaur-16-ol and kauran-16-ol were unique chemical components of the sunflower receptacle. A previous study showed that kaur-16-ene strongly inhibited cancer cells [13], so SEO may have the potential to inhibit cancer cells.

### 2.2. Antimicrobial Activities of SEO against Bacteria and Fungi

We evaluated antimicrobial activities of SEO against bacteria (*Escherichia coli*, *Pseudomonas aeruginosa* and *Staphylococcus aureus*) and fungi (*Saccharomyces cerevisiae* and *Candida albicans*) by MIC and minimum bactericidal concentration (MBC). Tetracycline hydrochloride was used as positive control against bacteria, and miconazole nitrate was used as positive control against fungi.

#### 2.2.1. Antibacterial Activities

Monomer mixtures contained α-terpineol, α-pinene, terpinolene and verbenone (m:m:m:m = 1:1:1:1). The antibacterial activities of SEO, monomer mixtures and monomers (α-pinene, verbenone, α-terpineol and terpinolene) were evaluated against *E. coli*, *P. aeruginosa* and *S. aureus* (Table 3 and Table A2). The results indicated that the MIC of SEO was 0.2 mg/mL against *P. aeruginosa* and *S. aureus*. The MIC of α-pinene was 6.4 mg/mL against *P. aeruginosa* and *S. aureus*. The MBC of α-terpineol against *P. aeruginosa* and *S. aureus* was 6.4 mg/mL. The MIC and MBC of α-terpineol against *E. coli* were 6.4 mg/mL. The MIC of monomer mixtures was 1.6 mg/mL against *P. aeruginosa* and *S. aureus*. A previous study showed that α-terpineol had antibacterial activity with a mechanism of changing the morphology of *E. coli* [14].

The MIC and MBC of SEO were lower than those of the monomer mixture. The MIC and MBC of SEO were lower than those of each monomer. The results showed that the antibacterial effect of SEO was better than that of the monomer mixture and each monomer. The results also implied that the other monomers of SEO may play an important role in the antibacterial effect.

Essential oils from other plants, such as essential oil of *Mentha citrata* Ehrh., *Artemisia annua* L. and *Citrus medica* L., were found to have antibacterial activities. Essential oil of *Mentha citrata* Ehrh. had significant antibacterial activity at 0.5 mg/mL against *S. aureus* [15]. The MIC of the essential oil from *Artemisia annua* L. against *S. aureus* was 10 mg/mL [16]. The essential oil from fruits of *Citrus medica* L. had the MIC of 1.56 mg/mL against *S. aureus* [17]. In this work, SEO was found to have antibacterial activity against *S. aureus* (MIC = 0.2 mg/mL). The smaller MIC value indicates the better antibacterial activity of essential oil. The results showed that SEO had better antibacterial activities than essential oils of other plants, such as *Mentha citrata* Ehrh., *Artemisia annua* L. and *Citrus medica* L.

*E. coli*, *S. aureus* and *P. aeruginosa* are the main pathogens causing some diseases affecting human health, such as diarrhea, urinary tract infections, skin infections and respiratory diseases [18,19,20]. In this study, SEO was found to have significant antibacterial properties against both Gram-positive and Gram-negative bacteria.

#### 2.2.2. Antifungal Activities

We evaluated the antifungal activities of SEO and monomers by using *S. cerevisiae* and *Candida albicans*. Table 4 and Table A3 summarized MIC and MBC of SEO and monomers. The MIC of SEO results implied that SEO had the highest antifungal activities against *S. cerevisiae* at 3.2 mg/mL. The antifungal activities of the major monomers of SEO, namely α-pinene, α-terpineol, verbenone and terpinolene, were tested separately. The results showed that α-pinene had the highest antifungal activity against *S. cerevisiae* (MIC = 0.8 mg/mL and MBC = 1.6 mg/mL) out of the SEO monomers. In addition, the α-pinene content of SEO was 26.00%. Previous studies also found that α-pinene had a great antifungal effect [21]. The other three main monomers, namely verbenone, terpinolene and α-terpineol, had strong antifungal activities, showing that α-pinene, α-terpineol, terpinolene and verbenone play primary roles in the antifungal activities of SEO against *S. cerevisiae* and *Candida albicans*.

Previous research proved that essential oil of Euodiae Fructus had antifungal effects against *Candida albicans* (MIC = 25.6 mg/mL) and essential oil of *Rosa* had antifungal activities against *Candida albicans* (MIC = 25 mg/mL) [22,23]. Here, SEO had antifungal activities (MIC = 3.2 mg/mL). The results of this experiment imply that SEO has better antifungal activities than essential oils of Euodiae Fructus and *Rosa*.

*Candida albicans* is the most common fungus, which found in the gastrointestinal and genitourinary tracts. The mucosa produced by *Candida albicans* may lead to infections in the gastrointestinal, oral and reproductive tracts when the human immune system was weak. There is currently no effective medicine for rapid clinical cure [24]. SEO and its monomers, such as α-pinene, α-terpineol and terpinolene, had good antifungal activities against *Candida albicans*.

### 2.3. Antioxidant Activity

#### 2.3.1. ABTS Radical Scavenging Activity

In the early stage, ABTS scavenging rate increased with the increasing concentration of SEO (Figure 1). When the concentrations of SEO were 0.1, 0.2, 0.4, 0.6, 0.8 and 1.0 mg/mL, the ABTS scavenging rates were 37%, 54%, 75%, 84%, 90% and 96%, respectively. The results of ABTS free radical scavenging showed that the free radical scavenging rate of SEO reached 96% when the concentration of SEO was 1.0 mg/mL, which implied that SEO had great free radical scavenging antioxidant activity.

#### 2.3.2. DPPH Radical Scavenging Activity

DPPH, as a stable free radical, has been widely used as a tool to evaluate free radical scavenging antioxidant activities. As shown in Figure 2, the DPPH free radical scavenging rate depended on the concentration of SEO. DPPH free radical scavenging rates were 15.90%, 22.40%, 39.01%, 65.94%, 75.25%, 92.57% and 100%, when the concentrations of SEO were 1, 2, 4, 6, 8, 9 and 10 mg/mL, respectively. It is significant that the DPPH free radical scavenging ability of SEO reached 100% when the concentration of SEO was 10 mg/mL, which is the same rate as that of the positive control trolox. The DPPH free radical scavenging activity results showed that SEO had good antioxidant activity. The antioxidant properties of terpenes and their derivatives were similar to those of phenolic compounds, which can scavenge free radicals by supplying hydrogen to hydrogen atoms [25]. It has been reported that α-pinene and α-celene react rapidly with peroxy radicals, resulting in rapid termination of the oxidative chain reaction and thereby reducing the number of reactive free radicals [26].

#### 2.3.3. Iron Ion Reduction Ability Analysis

Iron ion reduction ability and the concentration of SEO showed an approximately linear relationship when the concentration of SEO was increased from 0.1 to 8 mg/mL (Figure 3). The reducing power is related to the absorbance and increases with increasing absorbance. The results show that the iron ion reduction ability of SEO was related to the SEO concentration. The higher concentrations had better reducing abilities. When the concentration of SEO reached 8 mg/mL, the reduction ability was the greatest.

## 3. Materials and Methods

### 3.1. Chemical Reagents and Solvents

Tetracycline hydrochloride, miconazole nitrate, 2,2-diphenyl-1-picrylhydrazyl (DPPH) and (±)-6-hydroxy-2,5,7,8-tetramethylchromane-2-carboxylic acid (Trolox) were purchased from Source Leaf Biology Co. (Shanghai, China). Pentadecane, potassium persulfate and 2,2′-azino-bis(3-ethylbenzothiazoline-6-sulfonic acid) (ABTS) were purchased from Meilun Biology Co. (Dalian, China). Iron ion reduction capacity kit was purchased from Congyi Biology Co. (Shanghai, China). All other chemical reagents and solvents were of analytical grade. α-Pinene (99%) and α-terpineol (97%) were purchased from Shandong West Asia Chemical Industry Co. (Linyi City, China). Verbenone (97%) and terpinolene (98%) were purchased from Beijing Bailingwei Technology Co. (Beijing, China).

### 3.2. Plant Materials and Extraction of Essential Oil

The 4 kg sunflower receptacles were collected from Da’an City, Jilin Province (123°12′45″ E, 44°52′23″ N) in October 2019 and identified by Professor Shuwen Guan, School of Life Science, Jilin University. The samples were air-dried and then crushed into powder. In each experiment, 100 g powder was extracted by hydrodistillation in Clevenger-type apparatus for 8 h [27]. The experiments of extraction were repeated 30 times to provide enough SEO for analysis of biological activities. SEO was stored at 4 °C until analysis.

### 3.3. Analysis of Chemical Compositions of Essential Oil

The sample of SEO (10 μL) was diluted in n-hexane (10 μL) and analyzed with the Agilent 5975 (Agilent Technologies, Santa Clara, CA, USA). The separation was achieved using an HP-INNOWax capillary column (30 m × 0.25 mm i.d., 0.25 µm film thickness) (Agilent Technologies, Santa Clara, CA, USA). Helium was used as the carrier gas at a flow rate of 1 mL/min. Injector and detector temperatures were set at 250 and 280 °C, respectively. The oven was maintained at 60 °C for 3 min and then programmed to rise to 240 °C at 5 °C/min and held for 15 min at 240 °C. Electronic ion (EI) mode was set as 70 eV, mass spectra were recorded in the 50–550 amu range and the ion source temperature was 230 °C.

Retention indices of the separated compounds on the HP-INNOWax capillary column were determined on the basis of a homologous series of n-alkanes (C9–C30). The compounds of essential oil were identified on the basis of comparison of their retention indices and mass spectra with published data and computer matching with National Institute of Standards and Technology (NIST, 15.0) libraries provided with a computer controlling the GC-MS system. The relative proportions of SEO constituents were expressed as percentages obtained by peak area normalization, and all relative response factors were set as 1. Pentadecane was used as internal standard and α-pinene, verbenone, terpinolene and α-terpineol were used as external standards for GC-MS analysis.

### 3.4. Antimicrobial Effects

#### 3.4.1. Microbial Cultures of Three Bacterial Strains

Microbial cultures of three bacterial strains (*E. coli* (ATCC 25922), *P. aeruginosa* (ATCC 15442) and *S. aureus* (ATCC 25923)) were purchased from Huan Kai Microbiology Technology Co. (Guangzhou, China). Fungal strains (*Candida albicans* (ATCC 10231) and *S. cerevisiae* (ATCC 9080)) were also purchased from Huan Kai Microbiology Technology Co. (Guangzhou, China).

Bacteria strains were resuscitated in lysogeny broth (LB) solid medium. Then, they were transferred to LB liquid medium and incubated at 37 °C for 24 h. Yeast extract peptone dextrose medium and adenine (YPDA) solid medium was used as a medium for fungal recovery. Then, fungi were transferred to YPDA liquid medium and incubated at 26 °C for 48 h.

#### 3.4.2. Detection of Antimicrobial Activities

The MIC and MBC were measured by using a microplate reader (EL×800, BioTek, Winooski, VT, USA) [15]. Each monomer of essential oil (α-pinene, α-terpineol, terpinolene and verbenone) in the culture medium had nine different concentrations, namely 0.05, 0.1, 0.2, 0.4, 0.8, 1.6, 3.2, 6.4 and 12.8 mg/mL. Monomer mixture (with α-pinene, α-terpineol, terpinolene and verbenone) and SEO in the culture medium had nine different concentrations, namely 0.05, 0.1, 0.2, 0.4, 0.8, 1.6, 3.2, 6.4 and 12.8 mg/mL. A 200 μL mixed sample was added to each well of a 96-well microplate. The 200 μL mixed sample was mixed with 179 μL culture medium, 20 μL sample and 1 μL 2.0 × 10^6^ CFU/mL bacteria or 2.0 × 10^5^ CFU/mL fungi. The bacteria were cultured at 35–37 °C for 24 h, and the fungi were cultured at 25–26 °C for 48 h. The absorbance was measured by using the microplate reader at 600 nm. Tetracycline hydrochloride was used as positive control for the antibacterial experiments. Miconazole nitrate was used as positive control for the antifungal experiments. Dimethyl sulfoxide (DMSO) at 5% (*w/v*) and DMSO at 1% (*w/v*) were used as negative controls of antibacterial and antifungal experiments, respectively. The experiments were carried out in triplicate.

### 3.5. Determination of Antioxidant Activities

As a single antioxidant method is not able to accurately evaluate the antioxidant activity of SEO, ABTS, DPPH and iron ion reducing ability were used to evaluate the antioxidant activity of SEO.

#### 3.5.1. ABTS Radical Scavenging

ABTS radical scavenging activity was determined by the modified protocol from Kang [28]. The ABTS working solution was mixed with 2.6 mmol K_2_S_2_O_8_ and 7.4 mmol ABTS, which was incubated for 12 h at room temperature in the dark and diluted 40–45 times with ethanol. We added 0.5 mL sample to 2 mL ABTS working solution and incubated for 6 min at room temperature in the dark. The absorbance was measured at 734 nm. Trolox was used as a positive control.

The ABTS scavenging rate was determined by the following formula:$$\mathrm{ABTS}\;\mathrm{scavenging}\;\mathrm{rate}\;=\;[\;(A_{0}-A_{1})/A_{0}]\;\times \;100\%$$
where A_0_ is the absorbance of the negative control without SEO and A_1_ is the absorbance of the test sample with SEO.

#### 3.5.2. DPPH Radical Scavenging

DPPH radical scavenging activity was determined according to the modified protocol from Das [29]. Ethanol and DPPH were mixed to prepare 0.08 mmol/L DPPH solution, which was stored in the dark. Here, 1 mL sample and 3 mL DPPH solution were mixed and kept at room temperature for 30 min in the dark. The absorption value was measured at 517 nm. Anhydrous ethanol and Trolox were the negative and positive controls, respectively.

The DPPH radical scavenging capacity was determined by the following formula:$$\mathrm{DPPH}\;\mathrm{scavenging}\;\mathrm{rate}\;=\;[\;(A_{0}-A_{1})/A_{0}]\;\times \;100\%$$
where A_0_ is the absorbance of the negative control without the SEO and A_1_ is the absorbance of the test sample with SEO.

#### 3.5.3. Iron Ion Reducing Assay

The iron ion reducing ability was determined using Congyi Biology kit (Shanghai, China), according to the method of Prussian blue [30]. The antioxidant activity can change the ferric iron of potassium ferricyanide to ferrous ions. Which formed to Prussian blue.The material had a maximum absorption peak at 700 nm. A larger absorption value means a better antioxidant capacity of the sample.

### 3.6. Statistical Analysis

All the experiments were conducted with three replications. One-way analysis of variance (ANOVA) and the mean comparisons were performed on all antimicrobial and antioxidant data by using program SPSS 20.0 (IBM Corporation, Armonk, NY, USA). Duncan’s multiple range tests were used to calculate the mean values. Differences between mean values at *p* < 0.05 were considered significant.

## 4. Conclusions

The work presented in this paper represents the first study on the chemical components and biological activities of SEO. The yield of SEO was 0.42% (*v*/*w*) by hydrodistillation. SEO contained 68 chemical compounds, as determined by GC-MS analysis. The main components were terpenoids, including α-pinene and verbenone. Through in vitro antimicrobial experiments, SEO and the monomers (α-pinene, α-terpineol, terpinolene and verbenone) were shown to have great antimicrobial effects. The antioxidative abilities (ABTS, DPPH and iron ion reduction ability) were determined in vitro, proving that SEO has strong antioxidant effects. Therefore, SEO is worthy of further exploration due to its antibacterial and antioxidant potential.

## Figures and Tables

**Figure 1 molecules-25-05244-f001:**
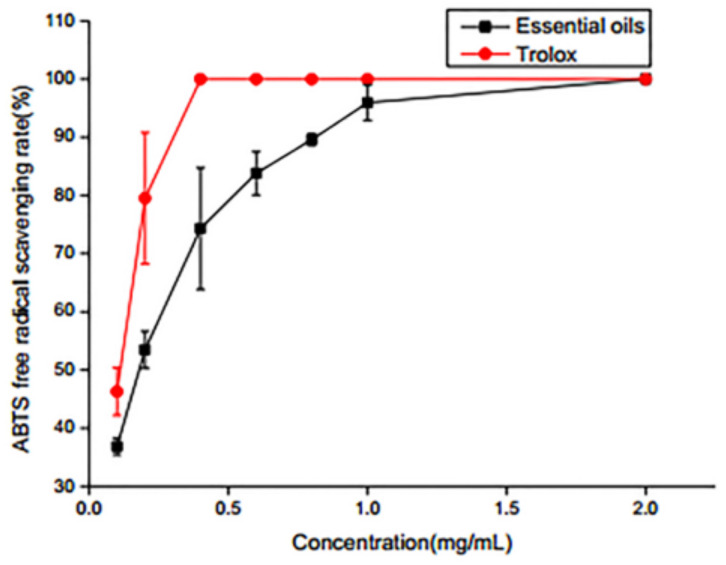
ABTS radical scavenging activity.

**Figure 2 molecules-25-05244-f002:**
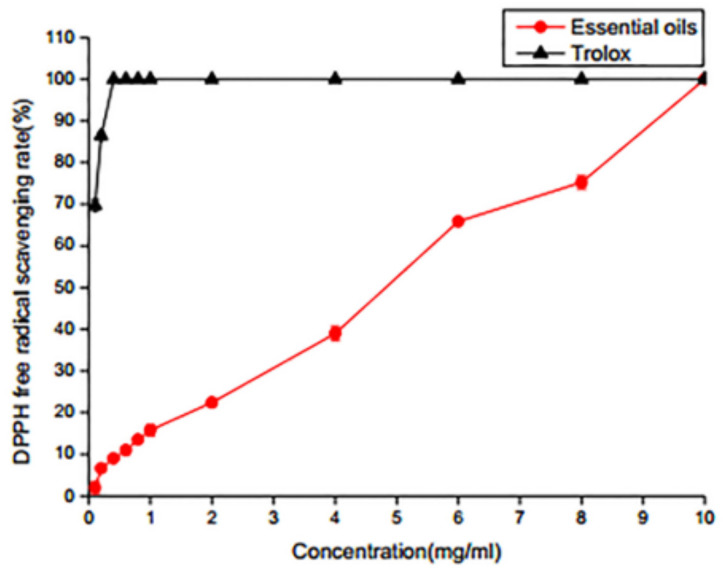
DPPH radical scavenging activity.

**Figure 3 molecules-25-05244-f003:**
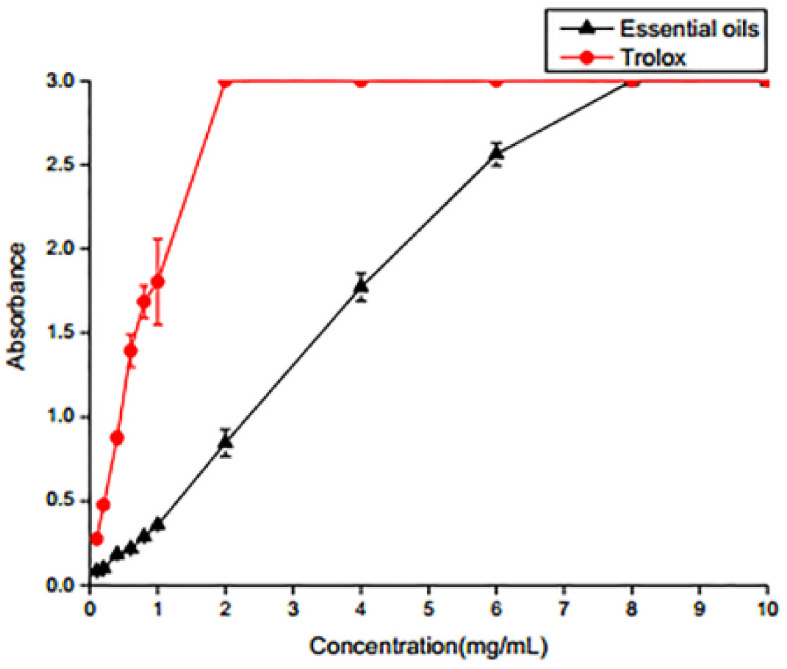
Iron ion reducing ability.

**Table 1 molecules-25-05244-t001:** Chemical composition of SEO.

NO.	Compound	Molecular Formula	RT ^a^	RI ^b^	MF ^c^	RMF ^d^	Content (%)	Identification ^e^
1	α-Pinene	C_10_H_16_	3.70	930	953	954	26.00	1,2,3
2	Camphene	C_10_H_16_	3.87	943	906	911	0.21	1,2
3	2,4-Thujadiene	C_10_H_14_	3.93	957	844	858	0.76	1,2
4	β-Terpinene	C_10_H_16_	4.26	964	876	918	0.15	1,2
5	l-β-Pinene	C_10_H_16_	4.31	969	926	935	0.38	1,2
6	Epoxycyclooctane	C_8_H_14_O	4.39	971	896	912	0.22	1,2
7	2,3-Dehydro-1,8-cineole	C_10_H_16_O	4.50	1041	791	835	0.33	1,2
8	1,3,8-*p*-Menthatriene	C_10_H_14_	4.56	1042	872	913	0.25	1,2
9	*E*,*E*-2,6-Dimethyl-1,3,5,7-octatetraene	C_10_H_14_	4.79	1049	902	915	0.27	1,2
10	4-Isopropenyltoluene	C_10_H_12_	4.87	1073	849	895	0.07	1,2
11	o-Cymene	C_10_H_14_	5.10	1079	929	939	0.62	1,2
12	Isosylvestrene	C_10_H_16_	5.27	1083	879	923	0.11	1,2
13	γ-Terpinen	C_10_H_16_	5.83	1101	908	918	0.18	1,2
14	*trans*-*p*-Mentha-2,8-dienol	C_10_H_16_O	5.99	1111	746	747	1.30	1,2
15	Berbenol	C_10_H_16_O	6.16	1117	824	830	0.13	1,2
16	Campholenal	C_10_H_16_O	6.24	1119	854	875	0.10	1,2
17	4-Isopropenyltoluene	C_10_H_12_	6.31	1123	919	940	0.33	1,2
18	Terpinolene	C_10_H_16_	6.40	1124	901	925	1.69	1,2,3
19	Benzyl ethyl carbinol	C_10_H_14_O	7.05	1131	761	799	0.35	1,2
20	l-Pinocarveol	C_10_H_16_O	7.24	1143	920	928	0.17	1,2
21	*cis*-Verbenol	C_10_H_16_O	7.32	1146	862	899	0.82	1,2
22	d-Verbenol	C_10_H_16_O	7.41	1149	917	919	4.11	1,2
23	Pinocarvone	C_10_H_14_O	7.54	1157	873	891	1.14	1,2
24	l-Terpinen-4-ol	C_10_H_18_O	8.00	1158	909	920	0.56	1,2
25	*p*-Cymen-8-ol	C_10_H_14_O	8.07	1160	906	920	0.61	1,2
26	Myrtenal	C_10_H_14_O	8.13	1169	927	933	0.07	1,2
27	α-Terpineol	C_10_H_18_O	8.19	1172	821	839	1.27	1,2,3
28	Verbenone	C_10_H_14_O	8.34	1198	908	917	7.40	1,2,3
29	*cis*-Carveol	C_10_H_16_O	8.76	1207	942	944	2.20	1,2
30	l-Carveol	C_10_H_16_O	8.95	1213	704	719	0.25	1,2
31	Carvol	C_10_H_14_O	9.02	1217	857	894	0.25	1,2
32	Hotrienol	C_10_H_16_O	9.10	1218	657	747	0.17	1,2
33	3,5-Diethylphenol	C_10_H_14_O	9.17	1219	762	799	0.16	1,2
34	*trans*-2-Caren-4-ol	C_10_H_16_O	9.38	1224	734	750	0.19	1,2
35	Bornyl acetate	C_12_H_20_O_2_	9.95	1259	887	892	0.72	1,2
36	(−)-*trans*-Pinocarvyl acetate	C_12_H_18_O_2_	10.16	1264	742	754	0.15	1,2
37	4-Vinylguaiacol	C_9_H_10_O_2_	10.25	1271	818	845	0.25	1,2
38	1,4-*p*-Menthadien-7-ol	C_10_H_16_O	10.65	1291	739	774	0.31	1,2
39	Aromadendrene, dehydro-	C_15_H_22_	12.42	1407	747	778	0.25	1,2
40	Calarene	C_15_H_24_	12.55	1412	903	930	5.27	1,2
41	4,5,9,10-dehydro-Isolongifolene	C_15_H_20_	12.91	1424	752	764	0.25	1,2
42	2-Tridecanone	C_13_H_26_O	13.39	1439	885	896	0.33	1,2
43	Bisabolene	C_15_H_24_	13.69	1450	896	912	0.52	1,2
44	Cadina-3,9-diene	C_15_H_24_	13.86	1456	835	851	0.30	1,2
45	Juniper camphor	C_15_H_26_O	14.20	1467	790	801	0.30	1,2
46	(−)-Spathulenol	C_15_H_24_O	14.52	1478	842	849	0.51	1,2
47	Caryophyllene oxide	C_15_H_24_O	14.59	1480	661	689	0.18	1,2
48	Isoaromadendrene epoxide	C_15_H_24_O	14.90	1490	775	791	0.70	1,2
49	*cis*-Lanceol	C_15_H_24_O	15.00	1494	749	805	0.43	1,2
50	3,3,5,6,7-Pentamethyl-1-indanone	C_14_H_18_O	15.35	1505	790	791	2.96	1,2
51	*Trans*-Longipinocarveol	C_15_H_24_O	16.09	1530	723	758	0.52	1,2
52	Dehydro-cyclolongifolene oxide	C_15_H_22_O	16.87	1556	740	752	4.81	1,2
53	Tetradecanoic acid	C_14_H_28_O_2_	17.20	1567	701	778	0.15	1,2
54	9-Hexadecenoic acid	C_16_H_30_O_2_	19.40	1701	661	669	0.22	1,2
55	Androst-2,16-diene	C_19_H_28_	19.62	1709	738	743	0.19	1,2
56	Phellopterin	C_17_H_16_O_5_	19.77	1773	766	786	2.17	1,2
57	Hexadecanoic acid	C_16_H_32_O_2_	19.84	1775	878	895	2.26	1,2
58	Manoyl oxide	C_20_H_34_O	20.12	1945	868	889	0.23	1,2
59	Androstane-3,11-diol	C_19_H_32_O_2_	20.81	1985	711	736	0.32	1,2
60	Methyl isopimarate	C_21_H_32_O_2_	21.35	1998	742	802	0.37	1,2
61	Linoleic acid	C_18_H_32_O_2_	21.57	2008	793	847	1.00	1,2
62	*trans*-Oleic acid	C_18_H_34_O_2_	21.65	2012	629	691	0.27	1,2
63	Kaur-16-ene	C_20_H_32_	22.23	2037	852	872	3.51	1,2
64	16-Kauran-16-ol	C_20_H_34_O	22.30	2040	870	888	1.65	1,2
75	Kauran-16-ol	C_20_H_34_O	22.43	2045	847	869	0.99	1,2
66	Cryptopinon	C_20_H_30_O	22.59	2052	769	798	0.29	1,2
67	Pimaric acid	C_20_H_30_O_2_	23.71	2101	739	741	3.09	1,2
68	Abietic acid	C_20_H_30_O_2_	24.26	2125	785	786	3.78	1,2
	Pentadecane ^f^							
	Total compounds						92.07	
	Oxygenated monoterpenes						54.65	
	Sesquiterpenoids						22.73	
	Others						14.69	

^a^ Peak time. ^b^ Retention indices relative to C9–C30 n-alkanes on the HP-INNOWax column. ^c^ Forward match. ^d^ Reverse match. ^e^ Methods of identification: 1, retention index; 2, mass spectrum; 3, co-injection with standard compound. ^f^ Internal standard.

**Table 2 molecules-25-05244-t002:** Comparison of the main components of essential oil from different sunflower parts (receptacle, capitula and pollen).

Compound	Receptacle		Capitula ^a^ [9]		Pollen ^b^ [10]	
	RI ^c^	Content (%)	RI	Content (%)	RI	Content (%)
α-Pinene	930	26.00	940	74.50	941	20.57
Verbenone	1198	7.40	1206	0.2	1205	1.52
Calarene	1412	5.27	-	-	-	-
Dehydro-cyclolongifolene oxide	1556	4.81	-	-	-	-
d-Verbenol	1149	4.11	-	-	-	-
Abietic acid	2125	3.78	-	-	-	-
Kaur-16-ene	2037	3.51	-	-	-	-
Pimaric acid	2101	3.09	-	-	-	-
3,3,5,6,7-Pentamethyl-1-Indanone	1505	2.96	-	-	-	-
Hexadecanoic acid	1775	2.26	-	-	-	-
*cis*-Carveol	1207	2.20	-	-	-	-
Phellopterin	1773	2.17	-	-	-	-
Terpinolene	1124	1.69	1089	0.2	-	-
16-Kaur-16-ol	2040	1.65	-	-	-	-
α-Terpineol	1172	1.27	1020	0.4	-	-
*trans*-*p*-Mentha-2,8-dienol	1111	1.30	1126	0.05	-	-
Pinocarvone	1157	1.14	1164	0.1	-	-
Linoleic acid	2008	1.00	-	-	-	-
Kauran-16-ol	2045	0.99	-	-	-	-
*cis*-Verbenol	1146	0.82	1143	0.7	-	-

^a^ The essential oil of sunflower capitula was extracted by water distillation in a Clevenger-type apparatus, and the components were identified by GC-MS. ^b^ The volatile components of sunflower pollen were identified by headspace solid-phase microextraction (HS-SPME)/GC-MS technique. ^c^ Retention indices. -, not detected with the same compounds and retention indices.

**Table 3 molecules-25-05244-t003:** Antibacterial activities of SEO and the main monomers.

Sample	*E. coli*		*P. aeruginosa*		*S. aureus*	
	MIC (mg/mL)	MBC (mg/mL)	MIC (mg/mL)	MBC (mg/mL)	MIC (mg/mL)	MBC (mg/mL)
SEO	6.4	12.8	0.2	0.2	0.2	0.4
α-Pinene	12.8	>12.8	6.4	12.8	6.4	6.4
α-Terpineol	6.4	6.4	3.2	6.4	6.4	6.4
Terpinolene	>12.8	>12.8	>12.8	>12.8	6.4	6.4
Verbenone	12.8	>12.8	>12.8	>12.8	3.2	3.2
Mixture ^a^	1.6	3.2	1.6	1.6	1.6	1.6
Tetracycline ^b^	ND	<0.05	ND	<0.05	ND	<0.05

ND, not detected. ^a^ Monomer mixture containing α-pinene, verbenone, terpinolene and α-terpineol (m:m:m:m=1:1:1:1). ^b^ Tetracycline hydrochloride was used as positive control for inhibiting bacteria.

**Table 4 molecules-25-05244-t004:** Antifungal activities of SEO and the main monomers.

Sample	*S. cerevisiae*		*Candida albicans*	
	MIC (mg/mL)	MBC (mg/mL)	MIC (mg/mL)	MBC (mg/mL)
SEO	3.2	3.2	6.4	12.8
α-Pinene	0.8	1.6	3.2	3.2
α-Terpineol	3.2	3.2	6.4	6.4
Terpinolene	1.6	1.6	1.6	1.6
Verbenone	12.8	12.8	6.4	12.8
Miconazole nitrate ^a^	ND	<0.05	ND	<0.05

^a^ Miconazole nitrate was used as positive control for inhibiting fungal. ND, not detected.

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
