# Peer review of "Chemical Composition and Antimicrobial and Antioxidant Activities of Essential Oil of Sunflower (Helianthus annuus L.) Receptacle"

_molecules, 2020, doi:10.3390/molecules25225244_

Round 1
Reviewer 1 Report
The present paper have its importance because describes the biological properties of an extract (essential oil) obtained from a by-product from sunflower. The manuscript is well designed and written, although I have some comments before recommending publication.
P1 L38 The sentence must be reformulated, because looks like it reffers to the present study and present plant matrix and is not true. Is a literature example.
P1 L41 Reformulate the sentence because ”terpenoid” is not a substance, but a class of compounds, so you can t refer to using singular (”was”)
P2 L45-46 It is not the previous study of the same authors, so please reformulate the sentence
Table 2 Attention to the design
Although S.cerevisiae is a single-cell fungus microorganism, I would prefer that you refer to it as ”yeast”
If the authors will make the corrections, I may recommend the publication.
My suggestion now is Minor revision
Author Response
Comment 1: P1 L38 The sentence must be reformulated, because looks like it refers to the present study and present plant matrix and is not true. Is a literature example.
Response: Thank you very much for your kind review and constructive suggestions. The revised sentence was showed in P1 line39-48 as followed " Sunflower roots, stems, leaves and seeds contained phenols, flavonoids and alkaloids [2]. In previous study, sunflower florets contained dietary fiber and phenolic acid [3]. Sunflower petals had triterpene glycosides, which had anti-inflammatory activity [4]. The ethanolic extract of sunflower seeds had the potential antidiabetic property in type 2 diabetes melitus [5]. While, the aqueous extract of sunflower seeds had considerable potential in reducing the asthma symptoms [6] and high antioxidant activity [7]. Many parts of sunflower have already been utilized for economic purpose. While sunflower receptacle always be discarded, because few studies focused on it and its commercial use has not been developed."
Comment 2: P1 L41 Reformulate the sentence because ”terpenoid” is not a substance, but a class of compounds, so you can t refer to using singular (“was”).
Response: I am agree with you that "terpenoid” is a class of compounds. So we changed "was" to "were". The revised sentence was showed in P2 line 52-53 as followed “The main chemical components of essential oil from industrial hemp (Cannabis sativa L.) and mediterranean herbs were terpenoids, which had great antioxidant and antibacterial activities [11, 12]".
Comment 3: P2 L45-46 It is not the previous study of the same authors, so please reformulate the sentence.
Response: you are right. The previous studies were not from the same author. So we separated the sentences with different authors. The revised sentences were showed in P2 line 53-63 as followed “ There were significant differences in the essential oil contents of different sunflower parts. The previous research showed that the content of α-pinene was higher in flowers (72.6%) than in leaves (28.6%) [13]. So it is important to compare the chemical components from different sunflower parts. Sunflower heads mainly consist of seeds, capitula, pollens, and receptacles. Sunflower seeds were rich in oleic acid [14]. While the main classes of chemical components of sunflower capitula and pollen were monoterpene hydrbons and sesquiterpene hydrocarbons, in which, the dominant volatile was α-pinene (20.0%) [15].”
Comments 4: Table 2 Attention to the design Although S.cerevisiae is a single-cell fungus microorganism, I would prefer that you refer to it as “yeast”.
Response: We changed " S.cerevisiae " to " S.cerevisiae (yeast) " in Table A3.
Comments 5: If the authors will make the corrections, I may recommend the publication.
My suggestion now is Minor revision.
Response: We made all the corrections and revised the manuscript, according to the reviewer comments.
Reviewer 2 Report
Remarks
1.The method of quantification, based on areas, is not safe because the chemical compounds are very different. Of course, the a-pinene is the predominant compound but the comparison or reference (e.g. lines 66-70), in my opinion is not correct. It is preferable to perform a semi-quantitative using an internal standard.
The above observation should be taken seriously, as you suggest applications in food, cosmetics, and medicines, where the specifications are very strict (lines 273 274).
- Line 19. Not yiled but yield.
- Line 20. Yield was 0.35% w / w or v/w? If the yield was expressed in w/w, how was the weight measured? Did you not have any losses? Usually the yield of the essential oil is calculated in v/w because the essential oil volume is determined immediately in Clevenger apparatus.
- Line 51. Receptacle of sunflower (SEO) because SEO is used for the first time in the main text.
- Many names in the tables are incorrect. The NIST library first displays the trunk of a chemical compound and then the branches. However, this is not in line with JUPAC. Please check the names.
- Line 191. How many grams of receptacle of sunflower were used?
- Materials and methods. How many times was each experimental procedure repeated?
- Table 1. Line 73. Mention that you used authentic, the correct term is standard, compounds. Which are they, the production companies and their purity? In addition, the use of these compounds is not mentioned in the identification of the compounds (lines 204-207).
- Table A1. Not α-pnene but α-pinene.
Author Response
Comment 1: The method of quantification, based on areas, is not safe because the chemical compounds are very different. Of course, the a-pinene is the predominant compound but the comparison or reference (e.g. lines 66-70), in my opinion is not correct. It is preferable to perform a semi-quantitative using an internal standard.
Response: Thank you very much for your professional suggestions and comments. They are helpful for improving the quality of the manuscript. We considered the comment seriously and we decided to reperform the GC-MS using Pentadecane as an internal standard and some compounds as external standards for semi-quantitative analysis.
The revised result and discussion part of the manuscript is as below in P2 line 80-89: " Pentadecane was used as an internal standard and some compounds (α-pinene, verbenone, terpinolene and α-terpineol) were used as external standards. Seventy three chemical components of SEO were identified by GC-MS analysis, accounting for 93.63% of the total content of essential oil, as shown in Table 1 and Figure A1. The major chemical composition was α-pinene (26.00%). The other main components were verbenone (7.40%), calarene (5.27%), kaur-16-ene (3.51%) and terpinolene (1.69%). Among of the 73 chemical compounds, most of them were belonged to monoterpenoids (55.80%), mainly including α-pinene, α-terpineol, verbenone and terpinolene. The content of sesquiterpene was 22.73% and others was 15.10%."
In the materials and methods part, we added the sentence “Pentadecane was used as internal standard and some compounds (α-pinene, verbenone, terpinolene and α-terpineol) were used as external standard for GC-MS analysis.” in P14 line 249-251.
Comment 2: The above observation should be taken seriously, as you suggest applications in food, cosmetics, and medicines, where the specifications are very strict (lines 273 -274).
Response: I am agree with you that the applications in food, cosmetics and medicines, where the specifications are very strict. The conclusion of this study should be rigorous. So we deleted the part of this sentence “in food, cosmetic and pharmaceutical industries.” The revised part of the manuscript is as below in P15 line 316-317: “Therefore, SEO could be used as natural antibacterial and antioxidants agents.”
Comment 3: Line 19. Not yiled but yield.
Response: I am agree with you. So we changed "yiled" to "yield". The revised word was showed in P1 line20.
Comment 4: Line 20. Yield was 0.35% w / w or v/w? If the yield was expressed in w/w, how was the weight measured? Did you not have any losses? Usually the yield of the essential oil is calculated in v/w because the essential oil volume is determined immediately in Clevenger apparatus.
Response: I agree with you. We used a electronic balance to measure the weight after drying the essential oil with sodium sulfate before, but there was a little amount of loss during the weighing process. So we repeat the experiment again, and we measure the volume of SEO directly and immediately in Clevenger apparatus as your good suggestion at this time. The result showed that the yield of SEO was 0.42% (v/w). The revised part of manuscript was showed in P1 line 21 and P2 line 74.
Comment 5: Line 51. Receptacle of sunflower (SEO) because SEO is used for the first time in the main text.
Response: Thanks very much for this comment. We changed “the essential oil of sunflower receptacle” to “SEO” in P2 line 67.
Comment 6: Many names in the tables are incorrect. The NIST library first displays the trunk of a chemical compound and then the branches. However, this is not in line with JUPAC. Please check the names.
Response: Thank you for your kind reminder. We have carefully checked all the names of the compounds and revised them in Table 1.
Comment 7: Line 191. How many grams of receptacle of sunflower were used?
Response: One hundred grams of air-dried sunflower receptacle was used in each experiment. The revised sentence was showed in P13 line 228-229 as followed ” The samples were air-dried and then crushed into powder. In each experiment, 100 g powder was extracted by hydro-distillation in Clevenger-type apparatus for 8 h”.
Comment 8: Materials and methods. How many times was each experimental procedure repeated?
Response: SEO extraction experiments were repeated 30 times. The experiments of biological activities were repeated three times. The revised sentence was showed in P13 line 231-232 as followed “The experiments of extraction were repeated 30 times to provide enough SEO for analysis of biological activities. "
Comment 9: Table1. Line 73. Mention that you used authentic, the correct term is standard, compounds. Which are they, the production companies and their purity? In addition, the use of these compounds is not mentioned in the identification of the compounds (lines 204-207).
Response: I am agree with you. So we changed "authentic” to "standard” in the note of Table 1. There were several compounds were used as standard external stand, such as α-pinene (99%), α-terpineol (97%), verbenone (97%) and terpinolene (98%). The revised sentence was showed in P13 line 221-224 as followed "α-Pinene (99%) and α-terpineol (97%) were purchased from Shandong West Asia Chemical Industry Co. (Linyi City, China). Verbenone (97%) and terpinolene (98%) were purchased from Beijing Bailingwei Technology Co. (Beijing, China)." These standard compounds as external standards for quantifying some compounds of SEO (α-pinene, verbenone, terpinolene and α-terpineol). The revised sentence was showed in P14 line 249-251 as followed " Pentadecane was used as internal standard and some compounds (α-pinene, verbenone, terpinolene and α-terpineol) were used as external standard for GC-MS analysis."
Comment 10: Table A1. Not α-pnene but α-pinene.
Response: We changed "α-pnene” to "α-pinene” in Table A2 (Table A1 was changed to Table A2 now).

Reviewer 3 Report
In the manuscript “Chemical Composition, Antimicrobial and Antioxidant Activities of Essential Oil of Sunflower (Helianthus annuus L.) Receptacle”, the authors investigated the chemical composition, antimicrobial and antioxidant activity of essential oil extract of Sunflower receptacle.
Although the manuscript is well-written, it is not clear the importance of investigating the chemical composition of sunflower`s receptacle. Is it due to potential bioactivity? Is it related to a unique or more varied metabolite constitution when compared to other parts of the plant? Is the receptacle somehow discarded of commercial use? If the main goal is to investigate the chemistry of this particular plant structure, the results should be compared with other parts. Which of the detected metabolites were previously identified in H. annuus? In which part and extract? Insert this information and the respective reference in the table. The authors should also provide more details about metabolite detection such as the MS matching score with the database.
Considering the extraction yield, how can the authors claim that “…the yield of essential oil from sunflower receptacle was higher than that from sunflower head” if the previous results are based on different procedures, especially time?
The in vitro biological assays showed that H. annuus essential oil extract contains potential antimicrobial and antioxidant properties. However, some comments should be considered
- A mixture of monoterpenes was more active against the bacteria than the individual components. Considering that they were tested at same concentration, how do the authors can justify that, then simply point to synergism? What would happen if the authors prepare different mixtures at various concentrations? It would be possible to evidence how each metabolite acts when mixed.
- The authors described the importance of searching for new antibiotics because “… these pathogens have shown resistance to various antibiotics…” but do the investigated strains have antibiotic resistance?
- There is no association between the in vitro antioxidant results and the chemical constitution of the essential oil. And why did the authors not test the mixture of terpenes?
Author Response
Comment 1:In the manuscript “Chemical Composition, Antimicrobial and Antioxidant Activities of Essential Oil of Sunflower (Helianthus annuus L.) Receptacle”, the authors investigated the chemical composition, antimicrobial and antioxidant activity of essential oil extract of Sunflower receptacle.
Comment 1: Although the manuscript is well-written, it is not clear the importance of investigating the chemical composition of sunflower`s receptacle. Is it due to potential bioactivity? Is it related to a unique or more varied metabolite constitution when compared to other parts of the plant? Is the receptacle somehow discarded of commercial use? If the main goal is to investigate the chemistry of this particular plant structure, the results should be compared with other parts. Which of the detected metabolites were previously identified in H. annuus? In which part and extract? Insert this information and the respective reference in the table. The authors should also provide more details about metabolite detection such as the MS matching score with the database. The composition comparison of various parts of sunflower is indicated in Table 2, and Table notes are added. As for your question about more MS data, I will illustrate it in the form of attached table.
Response: Thank you very much for your affirmation and professional suggestions. They are helpful for improving the quality of the manuscript. It is true that sunflower receptacle always be discarded as waste in the fields, because few studies focused on it and its commercial use has not been developed very well until now. If sunflower receptacle can be developed and utilized, it not only can increase farmers' income, but also can help us to protect the environment. So it is very important to investigate the chemical composition and the bioactivate of sunflower receptacle for developing the commercial use of sunflower receptacle.
In addition, we compared with the different parts of sunflower in the results and discussion part of manuscript, including sunflower receptacle in this study, and sunflower capitula and pollen in other previous studies. The results showed that the chemical composition of SEO in this study was more than that of sunflower capitula and pollen in previous studies.
Table 2 was revised in the manuscript. In addition, we added the notes of Table 2 and Table A1 for providing more detail information of GC-MS analysis. We considered the comments carefully and revised the manuscript as followed, according to the comments.
Firstly, we added the sentences in the introduction part of manuscript in P2 line 46-48 “Many parts of sunflower have already been utilized for economic purpose. While sunflower receptacle always be discarded, because few studies focused on it and its commercial use has not been developed. ”
Secondly, we added the sentences in the introduction part of manuscript in P2 line 53-60 “There were significant differences in the essential oil contents of different sunflower parts. The previous research showed that the content of α-pinene was higher in flowers (72.6%) than in leaves (28.6%) [13]. So it is important to compare the chemical components from different sunflower parts. Sunflower heads mainly consist of seeds, capitula, pollens, and receptacle. Sunflower seeds were rich in oleic acid [14]. While the main classes of chemical components of sunflower capitula and pollen were monoterpene hydrons and sesquiterpene hydrocarbons, in which, the dominant volatile was α-pinene (20.0%) [15].”
Thirdly, the results and discussion part of manuscript was revised as followed in P7 line 94-100 “The main chemical components were α-pinene and verbenone, which were all in sunflower receptacle, capitula and pollen, shown in Table 2. Among of the main chemical components, α-pinene was the highest content of essential oil. The same chemical components of essential oil were both in sunflower receptacle and capitula, such as terpinolene, α-terpineol, pinocarvone and cis-verbenol. However, the most of chemical components of essential oil from different sunflower parts (receptacle, capitula and pollen) were different, such as kaur-16-ene, 16-Kaur-16-ol and Kauran-16-ol were unique chemical components of sunflower receptacle.”
Finally, Table 2 was revised and the notes were added under Table 2 for providing more information in P8 line 113-116 “a The essential oil of sunflower capitula was extracted by water distilled in a Clevenger-type, and the components were identified by GC-MS. b The volatile components of sunflower pollen was identified by HS-SPME (headspace-solid phase microextraction)/GC/MS technique. c RI, Retention indices. -, Not detected with the same compounds and retention indices.”
Comment 2: Considering the extraction yield, how can the authors claim that “…the yield of essential oil from sunflower receptacle was higher than that from sunflower head” if the previous results are based on different procedures, especially time?
Response: I agree with your opinion. We deleted the sentence in manuscript. The revised sentence was showed in P2 line 78-79.
Comment 3: A mixture of monoterpenes was more active against the bacteria than the individual components. Considering that they were tested at same concentration, how do the authors can justify that, then simply point to synergism? What would happen if the authors prepare different mixtures at various concentrations? It would be possible to evidence how each metabolite acts when mixed.
Response: The mixture of monoterpenoids was prepared with α-pinene, α-terpineol, terpinolene and verbenone (m: m: m: m = 1:1:1:1). I am agree with you that the conclusion of synergy was not accurate. So we deleted this conclusion in the manuscript. The revised sentence was showed in P9 line 134-137.
Comment 4: The authors described the importance of searching for new antibiotics because “… these pathogens have shown resistance to various antibiotics…” but do the investigated strains have antibiotic resistance?
Response: The strains didn’t have antibiotic resistance in this study. The strains were all normal strains in this study, which were provided by biological companies. The results showed that SEO had antibacterial activities. The revised sentence was showed in P9 line134-135 as followed “The results implied that the other monomers of SEO may play an important role on the antibacterial effect.”
Comment 5: There is no association between the in vitro antioxidant results and the chemical constitution of the essential oil. And why did the authors not test the mixture of terpenes?
Response: The antioxidant activities of monomers and monomer mixtures were performed before in our work, but the results showed that the monomers and monomer mixtures had little antioxidant properties. Therefore, they were not written in the manuscript.
Round 2
Reviewer 2 Report
The authors' answers to the questions and comments are considered satisfactory.
Author Response
Thank you very much for your review and approval of our revised manuscript, which is helpful for improving our manuscript.
Reviewer 3 Report
Abstract
p.1, line 17: Please consider “In this work, we investigated the chemical composition, antimicrobial and antioxidant activities of SEO.”
p.1, line 20: Please use putatively identified instead of “identified”
p.1, line 21, Replace “and terpinolene” by “, terpinolene”
Introduction
p.1, line 38: Remove “While,”
p.1, line 40: Please use “Sunflower receptacle has always been discarded because of a lack of studies focusing on its commercial application.”
p.2, lines 42-44. Please remove the following sentences “Essential oil has good antioxidant and antibacterial activities and been widely used in food, cosmetics and pharmaceutical industries [9,10]. The main chemical components of essential oil from industrial hemp (Cannabis sativa L.) and mediterranean herbs were terpenoids, which had great antioxidant and antibacterial activities [11,12].” – Cannabis has nothing to do with this work. It will confuse the reader
p.2 lines 45-50: Please consider “There were significant differences in the essential oil contents of different sunflower parts. Previous research showed that α-pinene was higher in flowers (72.6%) than in leaves (28.6%) [13]. In capitula and pollen, α-pinene was found at 20.0% in addition to monoterpene and sesquiterpene hydrocarbons [15].“
Results and Discussion
please revise table 1: compounds 3 is 2,4-Thujadiene (not sure if it is natural product), compound 9 should be named as cosmene, according to literature compound 11 is a pesticide (DTXSID00342058), as well as compound 20 (DTXSID20967863) and compound 40 (DTXSID50341379). Not sure if compound 57 is a natural product. Compound 58 is a plasticizer.
p.5 line 119: The authors can`t support that “it implied that SEO can be used as excellent natural bacteriostatic agents.” since there is no study regarding safety or in vivo efficacy.
p.6 line 144: Similarly, the authors can`t write that “SEO could be used as natural fungicides for treating the diseases caused by Candida albicans.”
Conclusions
p.11, line 276: Please replace for “SEO could be explored due to its antibacterial and antioxidants potential.”
Author Response
Comment 1: p.1, line 17: Please consider “In this work, we investigated the chemical composition, antimicrobial and antioxidant activities of SEO.”
Response: Thank you very much for your kind review and constructive suggestions. We changed this sentence according to your good suggestion. The revised sentence was changed in P1 line 18 as followed “In this work, we investigated the chemical composition, antimicrobial and antioxidant activities of SEO."
Comment 2: p.1, line 20: Please use putatively identified instead of “identified”
Response: We changed “identified" to "putatively identified”. The revised word was showed in P1 line21.
Comment 3: p.1, line 21, Replace “and terpinolene” by “, terpinolene”
Response: Thank you very much. We changed “and terpinolene” to “terpinolene”. The revised word was showed in P1 line 23.
Introduction
Comments 4: p.1, line 38: Remove “While,”
Response: We removed “While”. The revised sentence was showed in P1 line 41 as followed “The aqueous extract of sunflower seeds had considerable potential in reducing the asthma symptoms [6] and high antioxidant activity [7]."
Comments 5: p.1, line 40: Please use “Sunflower receptacle has always been discarded because of a lack of studies focusing on its commercial application.”
Response: Thanks for the good suggestion. We used this sentence according to your advice. So we changed " Many parts of sunflower have already been utilized for economic purpose. While sunflower receptacle always be discarded, because few studies focused on it and its commercial use has not been developed. " to " Sunflower receptacle has always been discarded because of a lack of studies focusing on its commercial application. " in P1 line 42-45.
Comments 6:p.2, lines 42-44. Please remove the following sentences “Essential oil has good antioxidant and antibacterial activities and been widely used in food, cosmetics and pharmaceutical industries [9,10]. The main chemical components of essential oil from industrial hemp (Cannabis sativa L.) and mediterranean herbs were terpenoids, which had great antioxidant and antibacterial activities [11,12].” – Cannabis has nothing to do with this work. It will confuse the reader
Response: We deleted the sentences “Essential oil has good antioxidant and antibacterial activities and been widely used in food, cosmetics and pharmaceutical industries [9,10]. The main chemical components of essential oil from industrial hemp (Cannabis sativa L.) and mediterranean herbs were terpenoids, which had great antioxidant and antibacterial activities [11,12].” in line 47-50.
Comments 7:p.2 lines 45-50: Please consider “There were significant differences in the essential oil contents of different sunflower parts. Previous research showed that α-pinene was higher in flowers (72.6%) than in leaves (28.6%) [13]. In capitula and pollen, α-pinene was found at 20.0% in addition to monoterpene and sesquiterpene hydrocarbons [15].“
Response: That is a good idea. We changed this part "There were significant differences in the essential oil contents of different sunflower parts. The previous research showed that the content of α-pinene was higher in flowers (72.6%) than in leaves (28.6%) [13]. So it is important to compare the chemical components from different sunflower parts. Sunflower heads mainly consist of seeds, capitula, pollens, and receptacles. Sunflower seeds were rich in oleic acid [14]. While the main classes of chemical components of sunflower capitula and pollen were monoterpene hydrons and sesquiterpene hydrocarbons, in which, the dominant volatile was α-pinene (20.0%) [15]. " to " There were significant differences in the essential oil contents of different sunflower parts. Previous research showed that α-pinene was higher in flowers (72.6%) than in leaves (28.6%) [9]. In capitula and pollen, α-pinene was found at 20.0% in addition to monoterpene and sesquiterpene hydrocarbons [10]. " in line 50-53.
Results and Discussion
Comments 8: please revise table 1: compounds 3 is 2,4-Thujadiene (not sure if it is natural product), compound 9 should be named as cosmene, according to literature compound 11 is a pesticide (DTXSID00342058), as well as compound 20 (DTXSID20967863) and compound 40 (DTXSID50341379). Not sure if compound 57 is a natural product. Compound 58 is a plasticizer.
Response: We checked compound 3 (2,4-Thujadiene) from other literates (doi:10.1021/ie501301u.and doi:10.1021/jf0624244), which showed that compound 3 (2,4-Thujadiene) is natural product of essential oil. You are right that “compound 9 should be named as compound cosmene”. So we changed “2,6-Dimethyl-1,3,5,7-octatetraene,E,E- ” to “cosmene” . According to your comments, we deleted the three compounds (Compound 11, compound 20, compound 40) in Table 1, because they are pesticides. We didn’t find compound 57 in any literature, so we are also not sure whether it is a natural product or not. We deleted compound 57 for safe in Table 1. We deleted compound 58 in Table 1 too, because it is a pesticide.
Comments9: p.5 line 119: The authors can`t support that “it implied that SEO can be used as excellent natural bacteriostatic agents.” since there is no study regarding safety or in vivo efficacy.
Response: You are right. So we deleted this sentence “it implied that SEO can be used as excellent natural bacteriostatic agents.” in P6 line 134.
Comments 10: p.6 line 144: Similarly, the authors can`t write that “SEO could be used as natural fungicides for treating the diseases caused by Candida albicans.”
Response:You are right. So we deleted this sentence “it implied that SEO can be used as excellent natural bacteriostatic agents.” in P7 line 161.
Conclusions
Comments 11: p.11, line 276: Please replace for “SEO could be explored due to its antibacterial and antioxidants potential.”
Response: We changed this part as your advice. The revised sentence was showed as followed in P12 line 302 " SEO could be explored due to its antibacterial and antioxidants potential.”
Thanks again for your nice review.
This manuscript is a resubmission of an earlier submission. The following is a list of the peer review reports and author responses from that submission.